# Transition Metal Coordination Polymers with Trans-1,4-Cyclohexanedicarboxylate: Acidity-Controlled Synthesis, Structures and Properties

**DOI:** 10.3390/ma13020486

**Published:** 2020-01-19

**Authors:** Pavel A. Demakov, Artem S. Bogomyakov, Artem S. Urlukov, Aleksandra Yu. Andreeva, Denis G. Samsonenko, Danil N. Dybtsev, Vladimir P. Fedin

**Affiliations:** 1Nikolaev Institute of Inorganic Chemistry SB RAS, Novosibirsk 630090, Russia; demakov@niic.nsc.ru (P.A.D.); a.urlukov@g.nsu.ru (A.S.U.); andreeva@niic.nsc.ru (A.Y.A.); denis@niic.nsc.ru (D.G.S.); dan@niic.nsc.ru (D.N.D.); 2Department of Natural Sciences, Novosibirsk State University, Novosibirsk 630090, Russia; bus@tomo.nsc.ru; 3International Tomography Center SB RAS, Novosibirsk 630090, Russia

**Keywords:** metal–organic frameworks, coordination polymers, aliphatic ligands, synthesis, magnetochemical study, thermolysis, metal oxides

## Abstract

Five trans-1,4-cyclohexanedicarboxylate (chdc^2−^) metal–organic frameworks of transition metals were synthesized in aqueous systems. A careful control of pH, reaction temperature and solvent composition were shown to direct the crystallization of a particular compound. Isostructural [Co(H_2_O)_4_(chdc)]_n_ (**1**) and [Fe(H_2_O)_4_(chdc)]_n_ (**2**) consist of one-dimensional hydrogen-bonded chains. Compounds [Cd(H_2_O)(chdc)]_n_∙0.5nCH_3_CN (**3**), [Mn_4_(H_2_O)_3_(chdc)_4_]_n_ (**4**) and [Mn_2_(Hchdc)_2_(chdc)]_n_ (**5**) possess three-dimensional framework structures. The compounds **1**, **4** and **5** were further characterized by magnetochemical analysis, which reveals paramagnetic nature of these compounds. A presence of antiferromagnetic exchange at low temperatures is observed for **5** while the antiferromagnetic coupling in **4** is rather strong, even at ambient conditions. The thermal decompositions of **1**, **4** and **5** were investigated and the obtained metal oxide (cubic Co_3_O_4_ and MnO) samples were analyzed by X-ray diffraction and scanning electron microscopy.

## 1. Introduction

Metal–organic frameworks (MOFs) are one of the most fascinating family of solid-state materials because of their highly tunable compositions, structures and functional properties [1]. Lying on the crossing of fundamental inorganic/organic chemistry and development of novel materials, MOFs have become one of the most attractive research fields during the past two decades [2,3,4]. Typically, aromatic carboxylate ligands are used to construct porous coordination frameworks due to high rigidity of such linkers. For example, terephthalate (bdc^2−^) linker is reported in more than 2500 examples of metal–organic coordination polymers (CSD May 2019). On the other hand, its aliphatic counterpart, that is trans-1,4-cyclohexanedicarboxylate (chdc^2−^), is far less explored despite appropriate rigidity and availability. Among other reasons, a synthesis of such aliphatic-based MOFs is generally more challenging and requires a careful customization of reaction parameters, such as temperature, crystallization time, solvent composition, template role, coordination modulators, acidity/basicity of the reaction medium, etc. [5,6,7,8], which are often hard to rationalize. More efforts are required to prepare new aliphatic-based MOFs, however, the successful syntheses could lead to unusual crystal structures as a result of hydrophobic interactions between aliphatic moieties and their unique effect on the crystal packing [9,10,11]. An increased conformational lability of organic blocks provides ligand-driven structural transitions in the corresponding porous networks, including reversible breathing phenomena [12,13,14,15,16]. Also, a lower thermal stability of the aliphatic linkers allows for the introduction of defects into regular MOF structures [17], as well as synthesis of various metal oxide nanoparticles [18], and nanoporous carbon materials [19,20], etc.

According to the literature, only one cobalt [21], two cadmium [22,23] and one mixed-valence Fe(II/III) [24] MOF with the single bridge ligand trans-1,4-cyclohexanedicarboxylate have been prepared before. Recently, we also reported the modulation of crystallization of chdc-based MOFs by urotropine in organic solvents [25]. Motivated by a lack of development of aliphatic-based MOFs, we investigated the water-based reaction systems of trans-1,4-cyclohexanedicarboxylic acid (H_2_chdc) and transition metal cations (Co^2+^, Fe^2+^, Cd^2+^, Mn^2+^). The weak organic bases (1,4-diazabicyclo[2.2.2]octane, urotropine) were used to modulate and buffer the acidity of the reaction mixture for the optimal crystal growth conditions. Magnetic properties in the temperature range of 2–300 K were investigated. A controlled thermolysis of MOFs results in a formation of nanostructured metal oxides, the composition, structure and morphology of which were analyzed and discussed.

## 2. Materials and Methods

### 2.1. Materials

Trans-1,4-cyclohexanedicarboxylic acid (H_2_chdc, >97.0%) and 1,4-diazabicyclo[2.2.2]octane (DABCO, >98.0%) were purchased from TCI (Tokyo, Japan). Mn(ClO_4_)_2_∙6H_2_O (>99.0%) and urotropine (>99.0%) were purchased from Sigma Aldrich (St. Louis, MO, USA). HClO_4_ 65% water solution (reagent grade) and Fe powder (high-purity grade) were purchased from Reachem (Moscow, Russia). All reagents were used as purchased without further purification. Distilled water was used in all experiments.

### 2.2. Characterization Techniques

Infrared (IR) spectra in KBr pellets in the range 4000−400 cm^−1^ were recorded on a Bruker Scimitar FTS 2000 spectrometer (Billerica, MA, USA). Thermogravimetric (TG) analysis in the temperature range 30−600 °C was carried out using a Netzsch TG 209 F1 Iris instrument (Selb, Germany). The experiments were performed under He or Ar flow (80 cm^3^ min^−1^) at a 10 K min^−1^ heating rate. Elemental analysis was made on a EuroVector EA3000 analyzer (Pavia, Italy). Powder X-ray diffraction (PXRD) analysis was performed at room temperature on a Shimadzu XRD-7000 diffractometer (Cu-Kα radiation, λ = 1.54178 Å, Kyoto, Japan). pH of the solutions was measured by Anion 4100 pH meter (Novosibirsk, Russia). SEM images were made on TM-3000 Scanning Electron Microscope (Osaka, Japan). Typical conditions for the measurements were the following: 20.0 kV accelerating voltage, high vacuum pumping.

Investigation of magnetic properties of the compound **1** was performed on a Faraday balance (sensitivity ~3∙× 10^−7^ g). Measurements of the magnetic susceptibility were performed in the field of 9.7 kOe, the stabilization accuracy of the field strength was 2%. During measurements, the samples were placed in an inert helium atmosphere at the pressure of 5 torr. The magnetic susceptibility of the polycrystalline samples **4** and **5** was measured with a Quantum Design MPMS*XL* SQUID magnetometer (San Diego, CA, USA) in the temperature range 2–300 K with magnetic field of up to 5 kOe. None of the complexes exhibited any field dependence of molar magnetization at low temperatures. Diamagnetic corrections were made using the Pascal constants. The effective magnetic moment was calculated as µ_eff_(T) = [(3k/N_A_µ_B_^2^)χ*T*]^1/2^ ≈ (8χ*T*)^1/2^.

### 2.3. Synthesis

#### 2.3.1. Synthesis of [Co(H_2_O)_4_(chdc)]_n_ (**1**)

100 mg (0.34 mmol) of Co(NO_3_)_2_∙6H_2_O, 56 mg (0.33 mmol) of H_2_chdc and 24 mg (0.17 mmol) of urotropine were dissolved in 8.00 mL of H_2_O in a 10 mL glass flask and heated at 80 °C for 18 h. After the cooling at the room temperature, the crimson crystals were filtered off, washed with water and dried in air. Yield: 66 mg (67%). IR (KBr, cm^−1^): 3433 (br. m, ν_O-H_), 3276 (m), 2957 (m, ν_C-H_), 2947 (m, ν_C-H_), 2933 (m, ν_C-H_), 2865 (m, ν_C-H_), 2241 (w), 1552 (s, ν_COOas_), 1406 (s, ν_COOs_), 1363 (m), 1328 (m), 1292 (m), 1227 (w), 1213 (w), 1090 (w), 1048(w), 974 (m), 916 (m), 888 (w), 778 (s), 713 (m), 544 (m), 500 (m). Elemental analysis data calculated for [Co(H_2_O)_4_(chdc)]: C 31.9%, H 6.0%. Found: C 31.3%, H 5.8%.

#### 2.3.2. Synthesis of [Fe(H_2_O)_4_(chdc)]_n_ (**2**)

10 mg (0.18 mmol) of Fe powder were dispersed in 1.00 mL of H_2_O and 0.035 mL of 65% (0.56 mmol) HClO_4_ were added. After dissolution of iron, 28 mg (0.16 mmol) of H_2_chdc and 24 mg (0.17 mmol) of urotropine were added to the mixture in the glass ampoule. Then the ampoule was soldered, then sonicated for 10 min and heated at 80 °C for 18 h. After cooling at room temperature, the ampoule was opened and the colorless single crystals were selected for single crystal X-ray diffraction (SCXRD) analysis.

#### 2.3.3. Synthesis of [Cd(H_2_O)(chdc)]_n_∙0.5nCH_3_CN (**3**)

350 mg of Cd(NO_3_)_2_∙4H_2_O (1.13 mmol) were dissolved in 15.00 mL of acetonitrile. 250 mg (1.45 mmol) of H_2_chdc and 150 mg (1.34 mmol) of DABCO were dissolved in 15.00 mL of water. Then, the obtained solutions were mixed in a 50 mL Teflon vessel and heated at 100 °C for 24 h. After the cooling at the room temperature, the white precipitate was filtered off, washed with acetonitrile and dried in air. Yield: 192 mg (52%). IR (KBr, cm^−1^): 3534 (m, ν_O-H_), 3389 (br. m, ν_O-H_), 2936 (m, ν_C-H_), 2857 (m, ν_C-H_), 2257 (w), 2051 (w), 1643 (m), 1579 (s, ν_COOas_), 1450 (m), 1422 (s, ν_COOs_), 1386 (m), 1359 (m), 1328 (m), 1288 (m), 1270 (m), 1212 (m), 1043 (w), 975 (w), 928 (m), 902 (w), 887 (w), 809 (w), 790 (m), 758 (w), 738 (w), 691 (w), 675 (w), 537 (w), 470 (m). Elemental analysis data calculated for [Cd(H_2_O)(chdc)]∙0.5CH_3_CN: C 33.4%, H 4.2%, N 2.2%. Found: C 33.3%, H 4.0%, N 2.2%.

#### 2.3.4. Synthesis of [Mn_4_(H_2_O)_3_(chdc)_4_]_n_ (**4**)

300 mg (0.83 mmol) of Mn(ClO_4_)_2_∙6H_2_O, 120 mg (0.70 mmol) of H_2_chdc and 120 mg (0.86 mmol) of urotropine were dispersed in 5.00 mL of H_2_O in a 10 mL glass flask and sonicated for 5 min. Then the flask was heated at 80 °C for 18 h. After the cooling at the room temperature, the white precipitate was filtered off, washed several times with water and dried in air. Yield: 85 mg (51%). IR (KBr, cm^−1^): 3615 (w, ν_O-H_), 3572 (w, ν_O-H_), 3348 (br. m, ν_O-H_), 2932 (m, ν_C-H_), 2856 (m, ν_C-H_), 1623 (m), 1562 (s, ν_COOas_), 1448 (m), 1417 (s, ν_COOs_), 1388 (m), 1358 (m), 1327 (m), 1281 (m), 1210 (m), 1144 (m), 1109 (m), 1049 (w), 977 (w), 927 (m), 901 (m), 886 (w), 812 (w), 778 (m), 745 (m), 709 (m), 685 (m), 590 (w), 555 (m), 532 (m), 479 (s), 397 (m). Elemental analysis data calculated for [Mn_4_(H_2_O)_3_(chdc)_4_]: C 40.3%, H 4.8%. Found: C 40.6%, H 5.0%.

#### 2.3.5. Synthesis of [Mn_2_(Hchdc)_2_(chdc)]_n_ (**5**)

300 mg (0.83 mmol) of Mn(ClO_4_)_2_∙6H_2_O, 280 mg (1.63 mmol) of H_2_chdc and 120 mg (0.86 mmol) of urotropine were dispersed in 5.00 mL of H_2_O and 0.025 mL of 65% (0.40 mmol) HClO_4_ was added. Then the mixture was sonicated for 5 min and heated at 120 °C for 30 min in a screwed-cap glass vial to allow a partial evaporation of the reaction solution (ca. to 4 mL total volume). The crystallized white precipitate was hot-filtered and washed quickly with H_2_O to avoid the formation of the compound **4** impurity. After washing, the product was dried in air. Yield: 87 mg (34%). IR (KBr, cm^−1^): 3448 (br. w, ν_O-H_), 2928 (s, ν_C-H_), 2856 (s, ν_C-H_), 2659 (w), 2550 (w), 1672 (s), 1586 (s, ν_COOas_), 1540 (m), 1439 (m), 1409 (s, ν_COOs_), 1383 (m), 1362 (m), 1330 (m), 1273 (m), 1246 (m), 1214 (m), 1134 (w), 1110 (m), 1039 (w), 1024 (w), 977 (w), 953 (w), 916 (m), 882 (w), 801 (w), 770 (w), 741 (m), 722 (w), 711 (w), 682 (w), 543 (w), 529 (m), 472 (m), 404 (m). Elemental analysis data calculated for [Mn_2_(Hchdc)_2_(chdc)]: C 46.3%, H 5.1%. Found: C 46.4%, H 5.2%.

#### 2.3.6. Synthesis of Oxides

50–100 mg of **1**, **4** or **5** were placed in an open porcelain crucible and heated with 4 °C∙min^−1^ rate up to 600 °C in an oven, kept at 600 °C for 2 h and cooled to the room temperature with 4 °C∙min^−1^ cooling rate.

### 2.4. X-ray Crystallography

Diffraction data for single-crystals of **1**–**5** were obtained at 130 K on an automated Agilent Xcalibur diffractometer (Santa Clara, CA, USA) equipped with an area AtlasS2 detector (graphite monochromator, λ(MoKα) = 0.71073 Å, ω-scans). Integration, absorption correction, and determination of unit cell parameters were performed using the CrysAlisPro 1.171.38.46 program package [26]. The structures were solved by dual space algorithm (SHELXT [27]) and refined by the full-matrix least squares technique (SHELXL [28]) in the anisotropic approximation (except hydrogen atoms). Positions of hydrogen atoms of organic ligands were calculated geometrically and refined in the riding model. A crystal structure of **5** is solved in two alternatives: orthorhombic *Fdd*2 (**5o**) and monoclinic *P*2_1_ (**5m**). The crystallographic data and details of the structure refinements are summarized in Table A1. Cambridge Crystallographic Data Center (CCDC) numbers 1973662-1973668 contain the supplementary crystallographic data for this paper. These data can be obtained free of charge from The Cambridge Crystallographic Data Center at https://www.ccdc.cam.ac.uk/structures/.

## 3. Results and Discussion

### 3.1. Synthesis

Compound [Co(H_2_O)_4_(chdc)]_n_ (**1**) was obtained with 67% yield by heating (80 °C) of an aqueous solution of stoichiometric amount of cobalt (II) nitrate and *trans*-1,4-cyclohexanedicarboxylic acid in the presence of urotropine. The phase purity of **1** was confirmed by PXRD (Appendix A). The crystallization takes place in a mildly acidic conditions (pH_start_ = 4.5, pH_final_ = 5.0). In comparison, other cobalt-chdc compounds [Co_5_(OH)_8_(chdc)]_n_∙4nH_2_O reported earlier by Kurmoo et al. [21], was synthesized in hydrothermal conditions (170 °C) using an excess of NaOH to create strongly alkaline medium, which apparently facilitates a formation of hydroxyl-rich polynuclear {Co_5_(OH)_8_}_n_^2+^ building blocks. In moderately acidic synthetic conditions used herein, only mononuclear aqua-complexes were obtained. It should be noted that carrying out the synthesis of **1** without urotropine results in a recrystallization of H_2_chdc (See Appendix B, Figure A1 and Figure A2) after the cooling. On the contrary, an elevation of the basicity of the reaction system by either increasing the amount of urotropine (two times) or by a replacement of urotropine to the stronger base (NaOH) only lead to the formation of the unknown white powder. Details of the optimized synthetic methods for all compounds are summarized in Table 1.

Compound [Fe(H_2_O)_4_(chdc)]_n_ (**2**) was obtained by heating (80 °C) of aqueous solution of H_2_chdc, urotropine and Fe(ClO_4_)_2_, which was synthesized in situ by the dissolution of Fe in the solution of perchloric acid. Despite our attempts to avoid the oxidation of iron(II), the crystalline precipitate of **2** was always contaminated by hydrated ferric(III) oxide; therefore, the chemical composition and structure were established by a single-crystal X-ray analysis only. Urotropine is not included in the final coordination polymer, but is a necessary component of the reaction mixture, apparently as a pH modulator for the proper crystallization process. Remarkably, **2** is the first structurally characterized example of iron(II) 1,4-cychohexanedicarboxyalte reported in the literature, besides of one mixed-valence Fe(II/III) compound [Fe_2_O(chdc)_1.5_]_n_ [24]. **2** is isostructural to **1** and to [Ni(H_2_O)_4_(chdc)], which was reported by Kurmoo et al. [29] and Chen et al. [7].

Compound [Cd(H_2_O)(chdc)]_n_∙0.5nCH_3_CN (**3**) was synthesized in the solvothermal conditions by heating (100 °C) of the cadmium(II) nitrate, H_2_chdc and DABCO in the mixture of water and acetonitrile. The phase purity of **3** was confirmed by PXRD (Appendix A). The starting pH of the solution (pH_start_ = 4.8) is only slightly changed during the reaction (pH_final_ = 4.5). The synthesis of one-dimensional compound [Cd(H_2_O)_2_(chdc)]_n_, reported by Thirumurugan et al. [23], was carried out in water from cadmium acetate and H_2_chdc with piperidine as the acidity modulator, while [Cd_2_(DMF)(chdc)_2_]_n_, reported by Yoon and co-workers [22], was obtained from the organic medium (N,N-dimethylformamide). In our case, both water/acetonitrile mixture and DABCO appeared to be necessary for synthesis **3**, since the reaction in pure solvents did not result in any MOF precipitation except for recrystallization of H_2_chdc. Using one equivalent of NaOH instead of DABCO led to the formation of single crystals of the known compound [Cd(H_2_O)_2_(chdc)]. Oddly enough, no precipitate was formed if replacing 1,4-diazabicyclo[2.2.2]octane (DABCO, pK_b_ = 5.2) [30] to urotropine (pK_b_ = 9.5); therefore, in addition to being a very specific pH modulator, DABCO molecules probably act as an intermediate template facilitating the crystallization of **3**. The starting pH (pH_start_ = 4.8) is quite similar to the reported pH = 5…6 in the synthesis of [Cd(H_2_O)_2_(chdc)]_n_ [23], but the role of DABCO in our case is likely to fix pH at certain values to complement the templation effect.

New Mn-based MOFs [Mn_4_(H_2_O)_3_(chdc)_4_]_n_ (**4**) and [Mn_2_(Hchdc)_2_(chdc)]_n_ (**5**) were obtained in aqueous medium from the manganese(II) perchlorate, H_2_chdc and urotropine. Due to the similar chemical compositions, the simultaneous precipitation of these two compounds is hard to avoid. After optimization of the reaction conditions, the pure phase of **4** was obtained at lower acidity (pH_start_ = 4.9; pH_final_ = 5.2) while the compound **5** was isolated at higher acidity of the reaction solution (pH_start_ = 4.6; pH_final_ = 4.9) due to (i) greater molar ratio of chdc:Mn and (ii) additional presence of strong HClO_4_ acid. Such conditions are consistent with a more “acidic” nature of **5**, which contains a partially protonated Hchdc^−^ moieties, compared with **4**, where only fully deprotonated chdc^2−^ ligands are present. Importantly, the concentration of the urotropine in the synthesis of both **4** and **5** should not be lowered to avoid the formation of H_2_chdc crystals. The phase purity of the obtained **4** and **5** samples was confirmed by PXRD (Appendix A).

### 3.2. Structure Descriptions and Infrared Spectroscopy

An asymmetric unit of [Co(H_2_O)_4_(chdc)]_n_ (**1**) contains one Co atom, which has an octahedral coordination environment of four aqua ligands and two O atoms of two COO-groups, which are situated in trans positions (Figure 1). Co–OH_2_ distances are 2.0947(9) Å and 2.1168(10) Å, and Co–OCO distance is 2.0993(9) Å. The metal centers are bound by bidentate (e,e)-1,4-chdc ligands to form polymeric chains, which are packed into a dense crystal structure. It seems that both hydrogen bonding (interchain O_COO_–O_aqua_ distances are 2.762 Å for O(1)···O(12), 2.868 Å for O(1)···O(11), and 2.889 Å for O(2)···O(11)) and hydrophobic interactions between cyclohexane rings play an important role in the formation of the dense phase. The smallest interlayer H_CH_...H_CH_ distance is calculated to be 2.320 Å, which indicates a close linker-to-linker packing in **1**. There is also an intramolecular hydrogen bond between aqua ligand and O atom of the COO-group (O(2)···O(12) distance is 2.652 Å). **1** is isostructural to [Ni(H_2_O)_4_(chdc)]_n_ [7,29]. The synthesis of [Co(μ-H_2_O)(H_2_O)_2_(cis-chdc)]_n_ containing a cis-isomer of 1,4-cyclohexanedicarboxylate ligand is also reported [31].

Compound [Fe(H_2_O)_4_(chdc)]_n_ (**2**) is isostructural to **1**. Fe–OH_2_ distances are 2.1292(12) Å and 2.1524(12) Å, and Fe–OCO distance is 2.1076(11) Å. Interchain O_COO_···O_aqua_ distances are 2.753, 2.887, and 2.902 Å. The intramolecular O_COO_···O_aqua_ distance is 2.665 Å. Compound **2** is the first example of iron(II) cyclohexanedicarboxylate reported in the literature.

The asymmetric unit of the structure [Cd(H_2_O)(chdc)]_n_·0.5nCH_3_CN (**3**) contains two Cd atoms, two trans-1,4-cyclohexanedicarboxylate ligands and two molecules of coordinated water. Cd(1) has a distorted octahedral coordination environment of five carboxylate O atoms and one aqua ligand. Cd(2) has less distorted octahedral coordination environment, which also consists of five carboxylate O atoms and one aqua ligand. Cd–O distances are in range of 2.2224(16)–2.3860(17) Å. Cd(1) and Cd(2) are interconnected via three bridging COO-groups to form a binuclear unit {Cd_2_(μ-RCOO-κ^1^,κ^1^)_2_(μ-RCOO-κ^1^)}. The binuclear units are interconnected to form waved polymeric chains parallel to the c axis (Figure 2a). The chains are interconnected by (e,e)-1,4-cyclohexanedicarboxylate ligands in six directions to form a 3D porous network (Figure 2b). There are isolated cages in structure **3** (two cages per unit cell). The void volume in **3** is 14% (PLATON [32]) and the voids are occupied by the guest CH_3_CN molecules (Appendix A).

The asymmetric unit of [Mn_4_(H_2_O)_3_(chdc)_4_]_n_ (**4**) contains four Mn atoms, four trans-1,4-cyclohexanedicarboxylate ligands and three molecules of coordinated water. Mn(1) has a distorted octahedral coordination environment containing six O atoms of five COO-groups. Mn(2) and Mn(3) have less distorted octahedral coordination environment, which contains four O atoms of four COO-groups, one bridging and one terminal H_2_O molecule. Mn(4) has a slightly distorted square-pyramidal coordination environment containing five O atoms of five COO-groups. For octahedral Mn centers, the Mn–O distances are in the range of 2.0884(14)–2.3142(15) Å for water molecules and nonchelate COO groups. One Mn–O_COO,chelate_ distance is 2.3572(13) Å. For square-pyramidal Mn(4) center, Mn–O distances are 1.993(6)–2.219(7) Å and there is a long Mn(4)...O contact of 2.8047 Å. Mn atoms are interconnected via bridging carboxylate groups to form a decanuclear ring incorporating an endo-chdc^2−^ ligand (Figure 3a). Translating along the *a* and *b* axis, the decanuclear rings tile (004) plane to form polymeric layer with hexagonal topology (Figure 3b). Such a structure is unique for metal–organic frameworks, although there are reported examples of manganese or iron MOFs, which are constructed of endotemplated “honeycombs” with a different size or metal rings [10,11,33,34]. The layers alternate along the *c* axis, interconnecting by bridging chdc^2−^ ligands to form 3D metal–organic framework (see also Appendix A).

The crystal structure of the compound [Mn_2_(Hchdc)_2_(chdc)]_n_ (**5**) could be solved in two alternatives: orthorhombic *Fdd*2 (**5o**) and monoclinic *P*2_1_ (**5m**). Higher symmetry variant **5o** has less independent atoms, and demonstrates lower *R*_1_ value. However, in the structure **5o**, the chdc^2−^ ligand is disordered over two orientations around a two-fold rotary axis, whereas lower symmetry variant **5m** demonstrates no disordering in the structure, but is characterized by a higher *R*_1_ value. Since the topology of framework **5** is the same in both cases, a further description of the crystal structure is given for the higher symmetry variant **5o**.

The asymmetric unit of **5** contains one Mn(II) cation, which has an octahedral coordination environment consisting of six O atoms of six COO-groups (Figure 4a). Mn–O distances are in range 2.083(8)–2.211(3) Å for oxygen atoms of bridging COO groups and 2.325(4) Å for oxygen atom of monocoordinated carboxylate. Mn atoms are bridged through COO-groups to form polymeric chain running along the c axis. The chains are interconnected by (e,e)-1,4-cyclohexanedicarboxylate ligands to form a 3D metal–organic framework (Figure 4b). One of the 1,4-cyclohexanedicarboxylates is monoprotonated. The structure is densely packed and has no free solvent accessible volume.

Infrared spectra of the compounds **1**, **3**–**5** are typical for metal–carboxylate complexes. All spectra contain O–H stretching bands in the 3348–3448 cm^−1^ region, cyclohexane ring C(sp^3^)–H stretching bands in the 2856–2957 cm^−1^ region, COO antisymetric and symmetric stretching bands at 1552–1586 cm^−1^ and 1406–1422 cm^−1^, respectively. For **5**, the O–H band at 3448 cm^−1^ is very broad and weak and can be attributed to the presence of protonated COOH groups.

### 3.3. Magnetochemical Analysis

Temperature dependencies of the effective magnetic moment (µ_eff_) and inverse magnetic susceptibility for complex **1** are shown in Figure 5. The µ_eff_ value is 4.82 µ_B_ at 300 K and decreases to 4.69 µ_B_ at 80 K. The 1/χ dependence is linear in the temperature range of 80–300 K and obeys the Curie–Weiss law with the best fit parameters *С* = 2.99 K cm^3^ mol^−1^ and *θ* = −7 K. The values of the µ_eff_ at 300 K and Curie constant *C* are higher than the theoretical spin only for the values of 3.87 µ_B_ and 1.875 K·cm^3^/mol^−1^, respectively, which is typical for the Co(II) ion in the octahedral environment orbital contribution to the magnetic susceptibility [35,36]. Spin-orbit coupling causes decreasing of the µ_eff_ with lowering temperature.

The µ_eff_(T) and 1/χ(T) dependencies for **4** and **5** are presented in Figure 6. The µ_eff_ values at 300 K are 9.23 µ_B_ and 7.77 µ_B_ and decrease gradually with lowering temperature down to 1.28 µ_B_ and 3.4 µ_B_ at 2 K for complexes **4** and **5**, respectively. The 1/χ(T) dependencies are linear in the temperature range 30–300 K and obey the Curie–Weiss law with optimal values of Curie constant *C* and Weiss constant *θ* equal to 15.25 K·cm^3^/mol and −126 K for **4** and 8.033 K·cm^3^/mol and −19.5 K for **5**. The values of μ_eff_ at 300 K and the Curie constant *C* are lower than theoretical spin only ones 11.83 μ_B_ and 17.5 K·cm^−1^/mol for four noninteracting Mn (II) ions and 8.37 µ_B_ and 8.75 K·cm^3^/mol for two noninteracting Mn (II) ions for complexes **4** and **5**, respectively. A decrease of µ_eff_ with lowering temperature and negative values of the Weiss constant *θ* indicate the presence of antiferromagnetic exchange interactions between the spins of Mn (II) ions. It should be noted that the low μ_eff_ value for **4** at 300 K (9.23 µ_B_ instead of 11.83 μ_B_ theoretical value) indicates that antiferromagnetic coupling is strong even at room temperature. This behavior is reasonable due to the presence of four crystallographically independent Mn(II) ions and several possible ways of magnetic exchange within the polymeric layer with unique decanuclear Mn(II) rings.

### 3.4. Thermal Stability and Thermolysis

Thermogravimetric analyses of the compounds **1**, **3**–**5** were performed under He of Ar flow. Compound **1** shows weight loss ca. 24% occurring up to 120 °C, assigned to the full loss of coordinated water molecules (Appendix A). The first weight loss is followed by a plateau region ranging up to 460 °C with further heating leading to decomposition of the framework.

Compound **3** demonstrates weight loss ca. 12% at 160–180 °C, which corresponds to the loss of coordinated water and CH_3_CN guest molecules (Appendix A). This temperature is much higher than boiling point of acetonitrile (82 °С) and water, despite its coordinated nature. The reason for such pronounced thermal stability is isolated structure of pores (see the structure description in Section 2.2). The first weight loss in followed by a plateau region ranging up to 450 °C. Several attempts were performed to activate samples of **3** by heating at 80–100 °C in vacuum. It was shown that the crystallinity of activated samples is significantly reduced (Appendix A), and the samples do not show adsorption of N_2_ at 77 K and CO_2_ at 195 K.

Compound **4** demonstrates weight loss ca. 5% at 180–200 °C due to the loss of coordinated water molecules (Appendix A), and further decomposition at 530 °С. The thermal properties of **5** (Appendix A) are more complex and reveal multiple intermediate states due to consecutive liberation of organic ligands. The first weight loss at 250–270 °C (29%) leads to Mn_2_(chdc)_2_ intermediate (calculated—28%) apparently due to the escape of the H_2_chdc molecule. The second step at 330–340 °C (20%) leads to an unknown phase or a mixture, followed by pyrolysis, completed at 520 °C with the formation of MnO (21% of the residual sample mass vs. 23% in theory).

Due to the wide applicability of metal oxide nanoparticles and a number of advantages of MOF sources for pyrolysis [37,38,39,40,41,42], the compounds **1**, **4**, **5** were considered as the precursors for the synthesis of the corresponding oxides by thermolysis in air. The temperatures of the decomposition were chosen as 600 °C for all compounds, according to TG data. The thermolysis products were identified as cubic Co_3_O_4_ for 1 (Appendix A) and cubic MnO for **4**–**5** according to PXRD (Appendix A). The coherence scattering areas of crystalline oxide nanoparticles were estimated by the Scherrer equation. The corresponding numbers are 53.6(5) nm for Co_3_O_4_ derived from **1**, 43.5(4) nm for MnO derived from **4** and 55.7(9) nm for MnO derived from **5**. While the coherence scattering areas of MnO nanocrystallites obtained from **4** or **5** are quite similar, the macroscopic shape of the metal oxide aggregates is markedly different from each other. As revealed by scanning electron microscopy (SEM), the morphology of MnO aggregates is similar to that of the initial MOF crystals: rhomboidal blocks for **4** and sticks for **5**, respectively (Figure 7). Such a morphological “memory effect” is not uncommon and could provide a convenient way for the control of the texture of the oxide phase on a macroscopic level, which, in turn, will affect certain functional properties of the bulk material, such as particle density, surface area, catalytic activity, and chemical reactivity.

## 4. Conclusions

To summarize, five new metal–organic frameworks (MOFs) based on trans-1,4-cyclohexanedicarboxylate linker were synthesized and characterized by elemental analysis, IR spectroscopy, powder diffraction and X-ray single crystal analysis. Fine optimization of the reaction conditions, including the solvent composition, temperature and reaction mixture pH, was found to be crucial to achieve a phase pure product with high crystallinity.

Isostructural [Co(H_2_O)_4_(chdc)]_n_ (**1**) and [Fe(H_2_O)_4_(chdc)]_n_ (**2**) consist of one-dimensional hydrogen-bonded chains. Compounds [Cd(H_2_O)(chdc)]_n_∙0.5nCH_3_CN (**3**), [Mn_4_(H_2_O)_3_(chdc)_4_]_n_ (**4**) and [Mn_2_(Hchdc)_2_(chdc)]_n_ (**5**) possess three-dimensional framework structures.

Metal–organic frameworks with paramagnetic metal cations Co(II), Mn(II) were studied by magnetochemical and thermal analyses. For the compound [Co(H_2_O)_4_(chdc)]_n_ (**1**) with chain-like structure, a typical decrease of the magnetic moment µ_eff_ was observed with lowering temperature due to a spin-orbit coupling. For the metal–organic frameworks with Mn(II), a presence of antiferromagnetic exchange was revealed, which is especially strong in [Mn_4_(H_2_O)_3_(chdc)_4_]_n_ (**4**) even at room temperature due to different possible ways of magnetic exchange within the polymeric layer with unique decanuclear Mn(II) rings.

The thermolysis of the compounds based on Co(II) and Mn(II) in oxygen-containing atmosphere produces macroscopic aggregates made of nanosized oxide phases of cubic Co_3_O_4_ and MnO, respectively. These aggregates inherit the shape of the crystals of the initial MOFs, making possible a control of functional properties of the bulk material by varying the MOF precursor.

## Figures and Tables

**Figure 1 materials-13-00486-f001:**
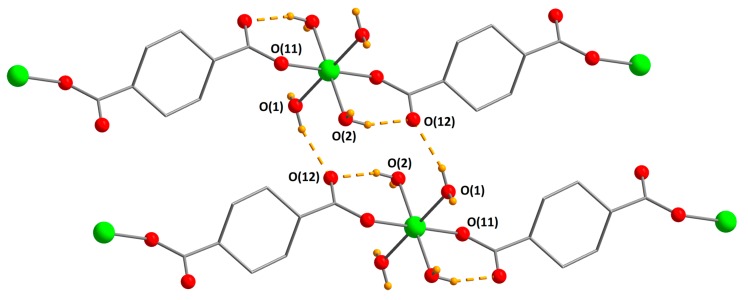
Packing of polymeric chains in **1** (CCDC 1973662). Metal atoms are green, O atoms are red, H atoms of aqua-ligands are orange. Hydrogen bonds are shown with dashed lines. H atoms of chdc^2−^ ligands are omitted.

**Figure 2 materials-13-00486-f002:**
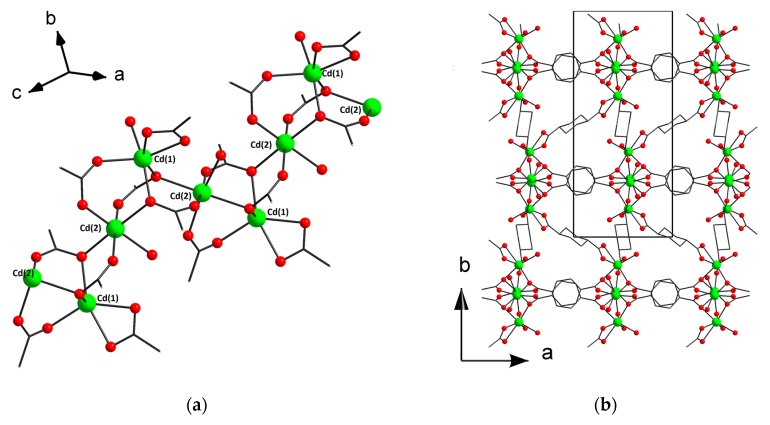
Fragment of polymeric chain (**a**) and 3D metal–organic framework (**b**) in **3** (CCDC 1973664). Hydrogen atoms and guest CH_3_CN molecules are omitted for clarity.

**Figure 3 materials-13-00486-f003:**
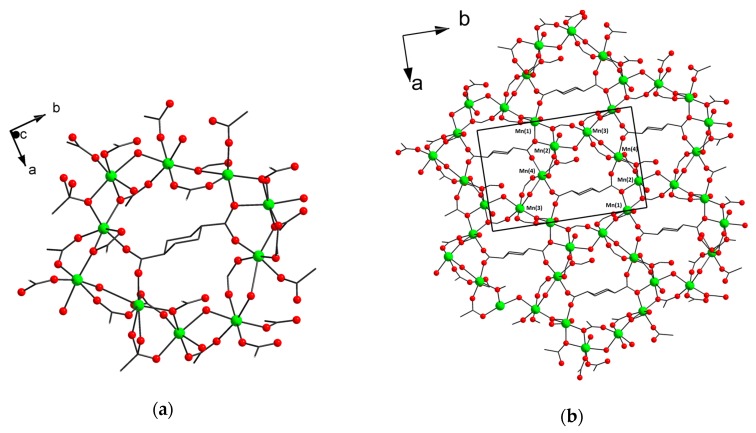
Decanuclear Mn(II) wheel with endo-chdc^2−^ ligand in **4** (CCDC 1973665) (**a**). Framework structure of **4** along the *c* axis (**b**).

**Figure 4 materials-13-00486-f004:**
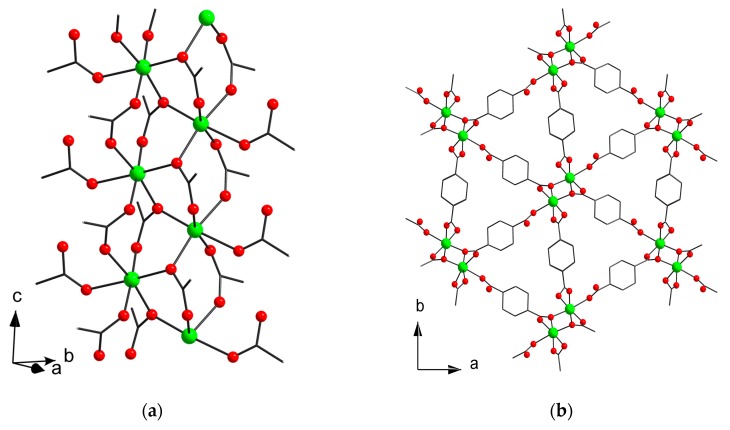
Fragment of polymeric chain in (**a**) and metal–organic framework (**b**) in **5o** (CCDC 1973666). Hydrogen atoms are omitted. Only one of possible orientations of chdc^2−^ ligand is shown.

**Figure 5 materials-13-00486-f005:**
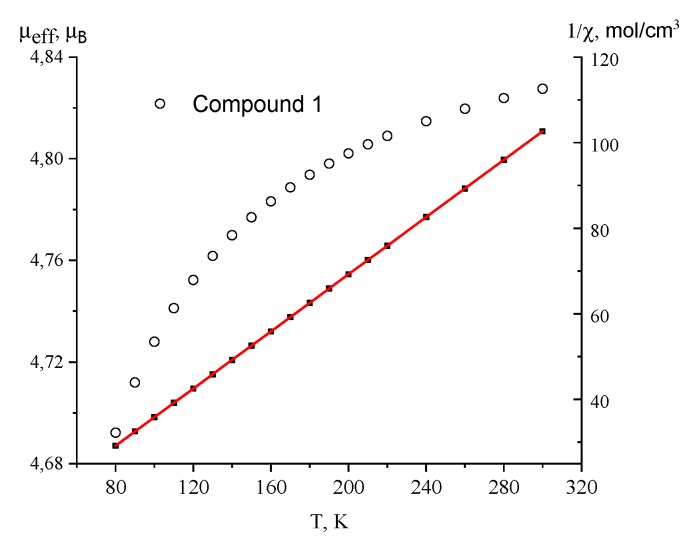
Temperature dependencies of the µ_eff_ (○) and 1/χ (■) for complex **1**. Solid line is a theoretical curve.

**Figure 6 materials-13-00486-f006:**
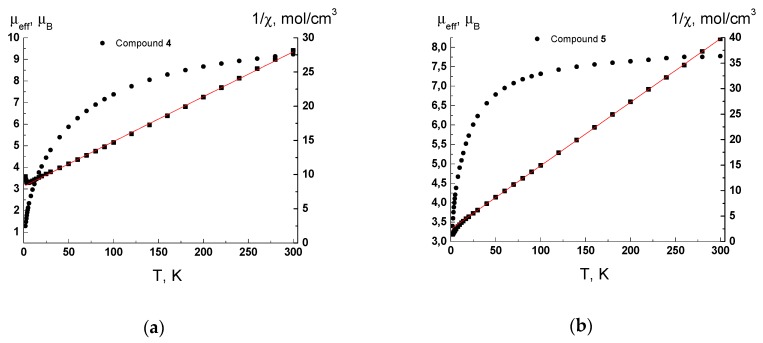
Temperature dependences of the µ_eff_ (●) and 1/χ (■) for complexes **4** (**a**) and **5** (**b**). Solid lines are theoretical curves.

**Figure 7 materials-13-00486-f007:**
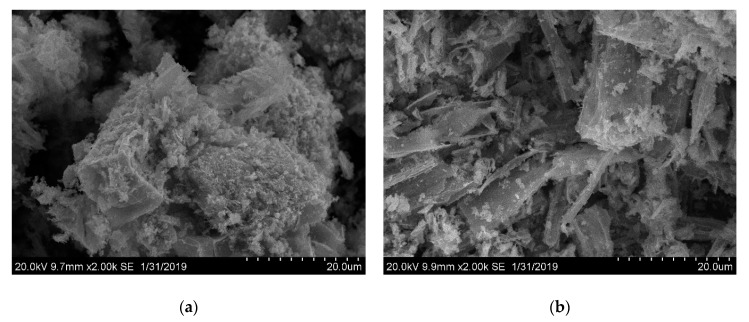
SEM images of MnO sample obtained by the oxidative thermolysis of **4** (**a**) and **5** (**b**).

**Table 1 materials-13-00486-t001:** Synthetic details for **1**–**5**.

Compound Number	Compound Formula	M^2+^: H_2_chdc: Base (Molar Ratio)	T, °C	pH_start_	pH_final_	Product at Higher pH	Product at Lower pH
**1**	[Co(H_2_O)_4_(chdc)]_n_	1: 1: 0.5	80	4.5	5.0	unknown	H_2_chdc
**2**	[Fe(H_2_O)_4_(chdc)]_n_	1: 1: 0.75	80	-	-	unknown	H_2_chdc
**3**	[Cd(H_2_O)(chdc)]_n_0.5nCH_3_CN	1: 1.3: 1.2	100	4.8	4.5	[Cd(H_2_O)_2_(chdc)]_n_	H_2_chdc
**4**	[Mn_4_(H_2_O)_3_(chdc)_4_]_n_	1: 0.8: 1	80	4.9	5.2	-	H_2_chdc
**5**	[Mn_2_(Hchdc)_2_(chdc)]_n_	1: 2: 1	80	4.6	4.9	-	H_2_chdc

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
