# Peer review of "Transition Metal Coordination Polymers with Trans-1,4-Cyclohexanedicarboxylate: Acidity-Controlled Synthesis, Structures and Properties"

_materials, 2020, doi:10.3390/ma13020486_

Round 1
Reviewer 1 Report
The paper by Fedin et. al. described transition metal coordination polymers with trans-1,4-cyclohexanedicarboxylate: acidity-controlled synthesis, structures and properties. Among them, compounds [Co(H2O)4(chdc)] (1) and [Fe(H2O)4(chdc)] (2) are isostructural and consist of one-dimensional hydrogen-bonded chains. Compounds [Cd(H2O)(chdc)]∙0.5CH3CN (3), [Mn4(H2O)3(chdc)4] (4) and [Mn2(Hchdc)2(chdc)] (5) possess three-dimensional framework structures. The compounds 1, 4 and 5 were further characterized by magnetochemical analysis. Furthermore, thermal decompositions of 1, 4 and 5 were investigated and the isolated metal oxide samples were analyzed by X-ray diffraction and scanning electron microscopy. Overall, this is a nice piece of work and can be accepted for publication in Materials with the following corrections.
All the compounds are polymer, therefore the molecular formula of [Co(H2O)4(chdc)] (1), [Fe(H2O)4(chdc)] (2), [Cd(H2O)(chdc)]∙0.5CH3CN (3), [Mn4(H2O)3(chdc)4] (4) and [Mn2(Hchdc)2(chdc)] (5) should be corrected as [Co(H2O)4(chdc)]n (1), [Fe(H2O)4(chdc)]n (2) {[Cd(H2O)(chdc)]0.5CH3CN}n (3), [Mn4(H2O)3(chdc)4]n (4) and [Mn2(Hchdc)2(chdc)]n (5). The author should assign characteristic peaks in the preparations section. And also describe IR spectral studies in the main text. Page 5, figure 1; the author should change the color of hydrogen atoms to avoid the similarity with the carbon atoms. Page 5, figure 1; the author should show the packing of polymeric chains of both the compounds 1 and 2 instead of common presentation. The presentation style of compounds 1-5 should be uniform and bold. For examples page 3, line 120, compound 4 should be compound 4; conditions for 1 should be conditions for 1; synthesis of 1 should be synthesis of 1. Page 4, line 151, “Is should be noted that” should be “It should be noted that” Some MOFs should be noted such as 10.1021/cg301573c, 10.1016/j.cclet.2017.09.050, 10.1021/cg100554u.
Reviewer 2 Report
Materials-692327
“Transition Metal Coordination Polymers with trans-1,4-cyclohexanedicarboxylate: Acidity-Controlled Synthesis, Structures and Properties”
P. Demakov, A. S. Bogomyakov, A. S. Urlukov, A. Y. Andreeva, D. G. Samsonenko, D. N. Dybtsev, V. Fedin
This paper describes the synthesis and characterization on five trans-1,4-cyclohexanedicarboxylate metal-organic frameworks of transition metals.
A careful control of pH, reaction temperature and solvent composition were shown to direct the crystallization of compound.
Magnetic properties were investigated. A controlled thermolysis of MOFs results in a formation of nanostructured metal oxides, which composition, structure and morphology were analyzed and discussed.
In general the work seems to be well done and deserve to be published in Materials. However, the authors should consider the References should be carefully revised.
Author Response
We appreciate and agree with the reviewer’s suggestion. The references have been carefully revised.
Reviewer 3 Report
This paper describes preparation of five trans-1,4-cyclohexanedicarboxylate metal-organic frameworks of transition metals. The authors examined the effect of the reaction conditions, including solvent composition, temperature and reaction mixture pH on the crystallinity of the products. They also studied the compounds with paramagnetic metal cations by magnetochemical and thermal analyses. I think both the experiments and the characterization of the products have no problem. I would like to accept this manuscript for publication in Materials.
The only comment I suggest is to describe what we can expect when we construct MOF with aliphatic dicarboxylate in the introduction section.
Reviewer 4 Report
The manuscript reports the synthesis of 5 MOFs based on trans-1,4-cyclohexanedicarboxylate, as well as investigation of their magnetic properties and thermal stability. Although the manuscript contains a lot of data, I feel that the authors have not gone all the way in drawing conclusions on their data, especially regarding the synthesis, and thus the way the manuscript is formulated now it seems like a list of experimental data. I therefore recommend thorough revision before acceptance.
In the abstract it is mentioned that “careful control of pH, reaction temperature and solvent composition were shown to direct the crystallization of a particular compound”. I feel that in the discussion part the authors could have drawn more conclusions based on their syntheses and when comparing the results obtained with different metals. Also, the optimization of the conditions is not discussed at all, although in the conclusions it is stated that fine tuning of the conditions is crucial to obtain phase pure materials. In the current state the discussion seems like a list of reaction conditions rather than insight into the factors that govern the synthesis. I suggest reformulation of this section. Also, it would be interesting to add the information of equivalents used of dabco/urotropine, since it is noted that the concentration is important for the formation of the desired structures.
I suggest adding data in table format throughout the manuscript to allow for comparison of the data obtained of the different structures, I think it would enhance the readability.
In the text throughout, I would suggest using the names of the MOFs (or abbreviated versions) instead of numbering compound 1, 2, 3, etc, which does not easily allow the reader to follow which MOFs is discussed.
Figure 1: which structure is shown, 1 or 2? Please give the CCDC number also in each figure caption. Please indicate the distances shown in the image.
As SEM imaging after pyrolysis is discussed, it would be interesting to see the images prior to pyrolysis of these two MOFs as well.
What conclusions can be drawn from the magnetic measurements? And the thermal stability? It would be of interest to the reader if more conclusions could be drawn from the data.
Some minor comments and typos:
Trans should be written in italics, please check the manuscript throughout.
32: omit the
35: replace prototype with counterpart
40: please reformulate the sentence to contain more information (e.g. that follows in the next sentences): “A lot more efforts are required to prepare new aliphatic-based MOFs, however, the successful results could be really interesting and intriguing”
46: omit a in “into a regular”
56: please reformulate as “, the composition, structure and morphology of which...are discussed”
2.1. materials: replace “received” with “purchased”, and vice versa. Details of the water quality that was used in the synthesis should be given. Write out and give here also the abbreviations used in the text later (H2chdc, dabco).
72: Can you give more information of the SEM imaging, such as kV used?
74: please check formatting of sensitivity
79: “none of the”, please add the
81: please check formatting of the formula
117: What is meant by “the mixture was sealed”? Can you estimate how much of the solution is evaporated in this time approximately?
151: replace “for 1” with “used herein”; “realized” by “obtained”, Is by It
251: carboxylate typo
255: densely
Round 2
Reviewer 4 Report
The authors have addressed most of the comments in the referee report. The manuscript is now in a better shape and can be recommended for publication.